# Microevolution and genomic epidemiology of the diphtheria-causing zoonotic pathogen *Corynebacterium ulcerans*

Chiara Crestani [1], Virginie Passet[1], Martin Rethoret-Pasty [1], Nora Zidane[1], Sylvie Brémont[1,2], Edgar Badell [1,2], Alexis Criscuolo [3] & Sylvain Brisse [1,2,3] ✉

*Corynebacterium ulcerans* is an emerging zoonotic pathogen which causes diphtheria-like infections. Although *C. ulcerans* is found in multiple domestic and wild animal species, most human cases are linked with pets. Our ability to decipher cross-host species transmission dynamics and to understand the emergence of clinically relevant clones (e.g., diphtheria toxin-positive) is currently hampered by a limited knowledge of *C. ulcerans* strain diversity and genome evolution. Here, we explore the genomic population structure and evolution of *C. ulcerans* with 582 isolates from diverse hosts and geographical locations. A newly developed core genome genotyping scheme captures the population structure of *C. ulcerans* both at deep and shallow phylogenetic levels, uncovering its main sublineages and offering high strain subtyping resolution for epidemiological surveillance. Additionally, we reveal the diversity and distribution of the diphtheria toxin gene (*tox*), and those of its associated mobile elements. Considering the entire *Corynebacterium diphtheriae* Species Complex, we find four diphtheria toxin families, five *tox*-prophage families, and a novel *tox*-carrying genetic element. We show that some toxin families are shared across *Corynebacterium* species, revealing *tox*-prophage cross-species transfer. Our work enhances knowledge on the ecology and evolution of *C. ulcerans* and provides a genomic framework for tracking the dissemination of emerging sublineages.

*C orynebacterium ulcerans*, a member of the *Corynebacterium diphtheriae* Species Complex (CdSC), causes diphtheria-like infections[1–3]. In the last decade, human cases have been on the rise in several countries[4–6]. In France, more *C. ulcerans* isolates are identified by the French national reference center (NRC) for diphtheria than *C. diphtheriae* itself, and the former are more often diphtheria toxin gene (*tox*) positive (56% and 32% in 2023, respectively). Although *C. ulcerans* is found in multiple domestic[7–11] and wild animal species[12–14], most human infections can be traced to contact with diseased or healthy pets (e.g., dogs and cats)[15–18].

Little is known about the origins of the *tox* gene in *C. ulcerans*, about its transmission dynamics among phylogenetic lineages, and its role in strain emergence. Besides, knowledge on the transmission of *C. ulcerans* among host species is scarce, especially with regard to wildlife, limiting our ability to control animal reservoirs and zoonotic transmissions.

Studies on the ecology, phylogeography, and small-scale evolution of *C. ulcerans* would benefit from a standardized genotyping system that could enable the definition of clinically relevant clones and track them through time and sources. Currently available genotyping

[1]Institut Pasteur, Université Paris Cité, Biodiversity and Epidemiology of Bacterial Pathogens, Paris, France. [2]Institut Pasteur, National Reference Center for Corynebacteria of the diphtheriae complex, Paris, France. [3]Institut Pasteur, Université Paris Cité, Biological Resource Center of Institut Pasteur, Paris, France. ✉e-mail: sylvain.brisse@pasteur.fr

methods for *C. ulcerans* include seven-gene multilocus sequence typing (MLST)[19,20], which has insufficient discriminatory power to detect transmission chains. High resolution can be achieved while leveraging the reproducibility and portability of MLST by applying this approach to the core genome (cg) of the species. Different from *C. diphtheriae*, for which a cgMLST scheme is already publicly available[21], no dedicated scheme currently exists for *C. ulcerans*, limiting the study of its population structure and the detection of epidemiological clusters.

*C. ulcerans* isolates can express the diphtheria toxin, a prominent virulence factor responsible for systemic clinical signs that worsen the prognosis[22]. The diversity of the *tox* gene has been previously assessed on limited genomic datasets[23–25]. In addition, even though the *tox* gene can be carried by prophages (as in *C. diphtheriae*) or by a pathogenicity island (PAI) described so far only in *C. ulcerans*[25–28], little is known about the diversity of the *tox*-carrying genetic elements and their distribution in the *C. ulcerans* population, as well as their association with specific *tox* variants.

In this work, a large genomic dataset was gathered, leveraging on the French NRC biocollection and international genomic sequence data, in order to: (i) define the population structure and microevolution of *C. ulcerans*; (ii) investigate the geographical and ecological spread of *C. ulcerans* lineages; and (iii) assess the *tox* gene diversity within *C. ulcerans* and the other species of the CdSC (including *C. diphtheriae*) and the distribution of *tox* variants and associated mobile elements within the population of *C. ulcerans*.

## Results and discussion
### Population structure of *C. ulcerans*
To define the genetic structure of *C. ulcerans*, a total of 434 genome assemblies were gathered. A cgMLST scheme was defined, which comprises 1628 highly conserved loci (Supplementary Information). To define clusters of isolates reflecting optimally the dissimilarity and phylogenetic discontinuities among *C. ulcerans* lineages, we analyzed the consistency properties of single-linkage clustering groups created based on a range of cgMLST allelic difference thresholds. A first threshold (belonging to an optimal plateau of the silhouette coefficient values; Supplementary Fig. S1) was selected at 194/1628 (11.9% allelic differences), resulting in 42 clusters defined as clonal groups (CG). A second threshold at 940/1628 (57.7% allelic differences) was chosen based on the combination of the consistency (silhouette) value and the observed distribution of allelic mismatch proportions (Supplementary Fig. S1), which partitioned the cgMLST profiles into 17 clusters defined here as sublineages (SL). To maximize the interpretability of the new CG and SL nomenclature, we attributed to these groups numerical identifiers inherited from the existing seven-gene MLST nomenclature (Supplementary Fig. S2; Supplementary Information), using a previously described inheritance algorithm[29].

The phylogenetic structure of the *C. ulcerans* population (see our Microreact project for an interactive exploration and more details) shows a few deep branches and several subtrees of highly similar genomes (Fig. 1), which is indicative of clonal expansions (e.g., CG331 or CG339, see below). These subtrees correspond in a few cases to documented outbreaks (e.g., CG325). In this aspect, *C. ulcerans* is strikingly different from *C. diphtheriae*, whose population structure shows deep branching of multiple (hundreds) of sublineages, and where no sublineage appears as being highly predominant[21,30]. This suggests the existence of a larger genetic pool of *C. diphtheriae* compared to *C. ulcerans*, even though the former is almost entirely restricted to humans. Sampling and sequencing more wild animal isolates could reveal currently hidden diversity of *C. ulcerans*, perhaps including sublineages that are not pathogenic for humans and hence currently not captured by diphtheria surveillance systems.

### Host and geographic distributions of *C. ulcerans* sublineages and clonal groups
Two major sublineages were uncovered (Fig. 1): SL325 (*n* = 211 isolates, 36.3%) and SL331 (*n* = 205, 35.2%). Most of the isolates belonging to these sublineages are *tox* gene-positive (*tox*+; 90%), with the *tox* gene being associated either with prophages (SL325) or with a PAI (SL331). These *tox* alleles are considered functional, as no predicted translated protein contained frameshift mutations or insertions/deletions leading to premature stop codons, and as most tested isolates were Elek-positive. In contrast, the remaining sublineages are, for the most part, *tox* gene-negative (*tox*–; Fig. 1), revealing that the true diphtheria-like causing *C. ulcerans* are SL331 and SL325.

Regarding their geographic distribution, only two sublineages were detected in more than one continent: SL325 (Europe: 95.7%, Asia: 3.8%, Africa: 0.5%) and SL339 (Europe: 93.8%, Africa: 4.7%, South America: 1.6%). All other sublineages, including SL331, are restricted to one continent (Europe, *n* = 13; South America, *n* = 2), likely because of their current lower sample sizes rather than a lack of large geographic distribution.

Clonal groups are finer-scale subdivisions of the population than sublineages. Whereas some sublineages comprise only one (major) clonal group (e.g., CG339 within SL339; Fig. 1), others are more heterogeneous, including the two *tox*+ SL325 and SL331. Important genomic and biological characteristics differed among clonal groups of given sublineages, as described below.

Four clonal groups were highly represented: CG325 (*tox*+), CG331 (*tox*+), CG339 (*tox*–), and CG583 (*tox*+, a member of SL325). CG325 isolates (*n* = 109) belong to an outbreak cluster among a group of dogs (2021–2022), and they all carry the *tox* gene (allele tox_49) in a single gene contig (orphan toxin, see below). CG339 (*n* = 64) is the most common *tox*– clonal group within *C. ulcerans*, and all its isolates are *tox*– (Fig. 1). This clonal group was first reported in 2015, and is mostly isolated from animals sampled in Europe, South America, and Africa.

In contrast, CG331 isolates (*n* = 96; isolated both from humans and animals) are primarily *tox*+ (Figs. 1 and 2A). Interestingly, this clonal group, first reported in 2008 (Fig. 2C) and geographically limited to Europe (France, Belgium, Germany, UK), is associated with several *tox* alleles (Fig. 2A) despite being genetically highly homogeneous. In CG331 genome assemblies, the *tox* gene is always carried by a PAI (except for allele tox_3, which was carried by a prophage).

CG583 (*n* = 67; Fig. 2B), the second most prominent clonal group within SL325, is common both in humans and pets (Fig. 1). In our dataset, CG583 is restricted to Europe (France, Belgium, Germany) and its first available isolate dates back to 1980 (Fig. 2C). Different from other major clonal groups of *C. ulcerans*, which are either fully *tox*+ or fully *tox*– (Fig. 1), CG583 isolates are variably *tox*– (31.3%, *n* = 21) or *tox*+ (68.7%, *n* = 46), in which case the *tox* gene is carried uniquely by a prophage. As *tox*– isolates are interspersed with *tox*+ ones along our phylogenetic classification, several losses and/or acquisitions by horizontal gene transfer must have occurred (Fig. 2B); this phylogenetic pattern suggests that these prophages excise or invade frequently the isolates of CG583. A similar situation can also be observed within CG543, a smaller clonal group of SL325 (Fig. 1).

Despite the bias towards isolates originating from Europe (see "Limitations" section), clonal groups comprising isolates from different countries (some being far from Europe, e.g., Japan) do not show deep branching that separate isolates from distinct geographical origins (e.g., CG328, CG337, CG514; Fig. 1); hence, major clonal groups found in France might be also represented in other countries. Moreover, all major clonal groups are associated with three or more host species, including humans, domestic and wild animals, underlying their zoonotic character. Future genomic studies of *C. ulcerans* from

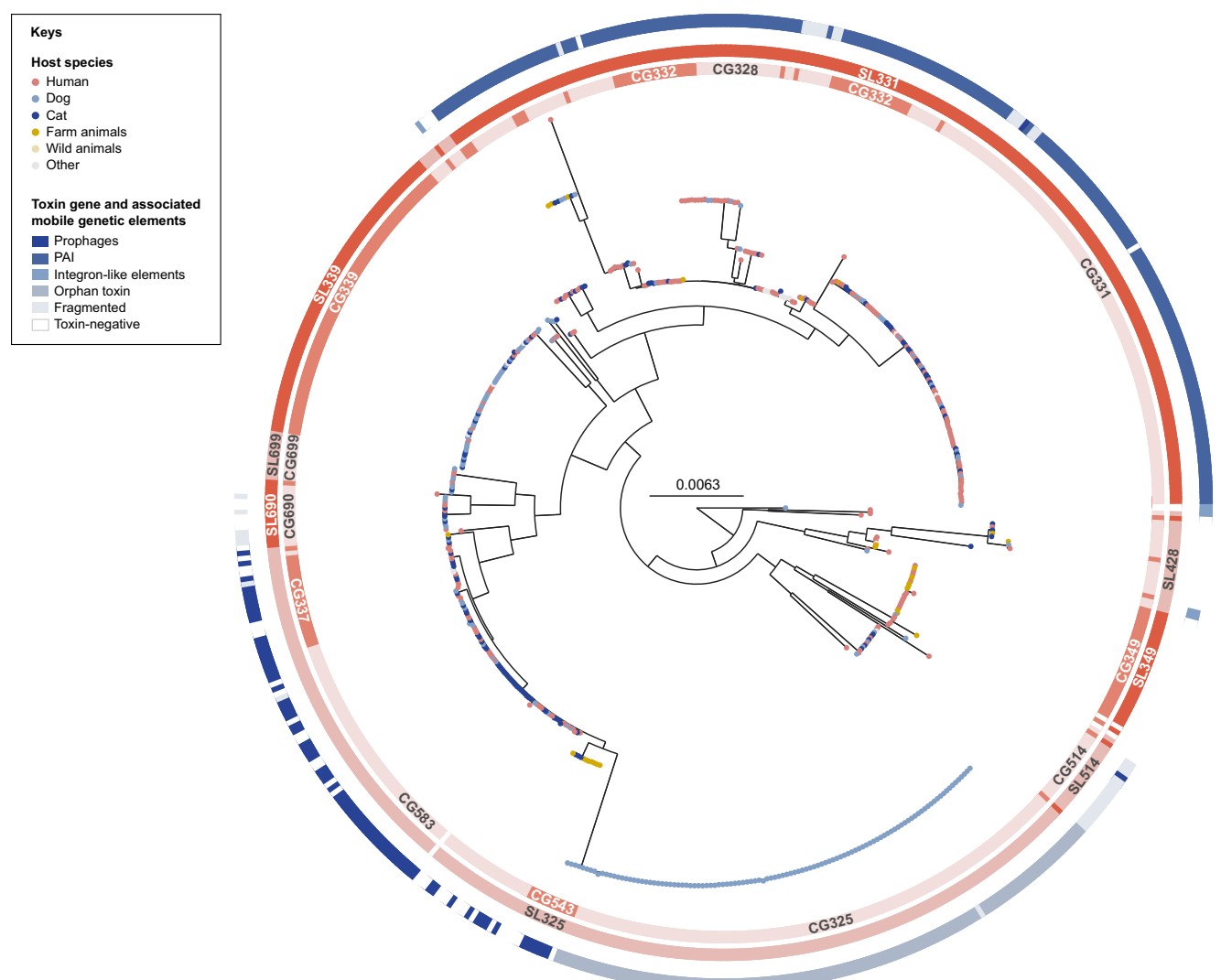

**Fig. 1 | Phylogenetic tree of 582 *Corynebacterium ulcerans* genomes from human and animal hosts.** This maximum-likelihood tree was inferred from the core gene set and rooted using a *Corynebacterium pseudotuberculosis* outgroup (collapsed). Colored leaves indicate the host species (human, dog, cat) or host group of origin (farm animals, wild animals). External strips show clonal groups (CG) and sublineages (SL), with their respective number inherited from the seven-gene MLST nomenclature for the most important ones (with white standing for not assigned). For all *tox*+ isolates, the associated mobile genetic element (carrying the *tox* gene) is shown in the outermost strip, when its identification was possible (e.g., except for genomes that were too fragmented).

other countries will likely enhance its overall reported genetic diversity.

## cgMLST variation within case clusters *versus* sporadic isolates

Tracking strain transmission of single genotypes across hosts is an important goal of genomic epidemiology. To calibrate cgMLST profile fluctuation with respect to epidemiological events, the cgMLST variation was evaluated within groups of isolates reported as epidemiologically related (i.e., a described cluster of cases, here named case clusters) and compared to variation among sporadic isolates (i.e., isolates considered as being epidemiologically unrelated based on available provenance information). Twenty case clusters were identified (Supplementary Table S1), six of which from the French NRC data and 14 of which from publicly available data from other countries (Belgium, Spain, Germany, and the UK). Four clusters (reported in Belgium) were not analyzed as only one genome was available for each. The remaining case clusters mostly involved humans and their contact pets, and they comprised isolates from various Sequence Types (STs) with different *tox* variants (Supplementary Table S1). We found a range of 0–19

cgMLST allelic differences among case clusters, in contrast to a mean variation of >1100 when considering the entire dataset (Fig. 3A, B).

## cgMLST genetic clusters reveal cryptic transmission of *C. ulcerans*

To date, the transmission dynamics of *C. ulcerans* have been poorly documented, and the role of wildlife as a reservoir of human and domestic animal infections is still unclear. Genomic surveillance can indirectly inform on transmission, by inference of the geographic and host distribution of closely related isolates. A threshold of 25 cgMLST allelic mismatches (selected from the distribution of the pairwise allelic mismatches in the overall dataset; Fig. 3A) was used to classify the 582 *C. ulcerans* genomes into 351 single-linkage clusters, named genetic clusters. These capture correctly all reported epidemiological clusters of infection (case clusters; *n* = 20) and enable the definition of cryptic clusters (*n* = 38) for which no prior information on epidemiological links is available.

Cryptic clusters (Fig. 3B) show a higher genetic heterogeneity compared to case clusters (Fig. 3C), extending to just under 70 allelic

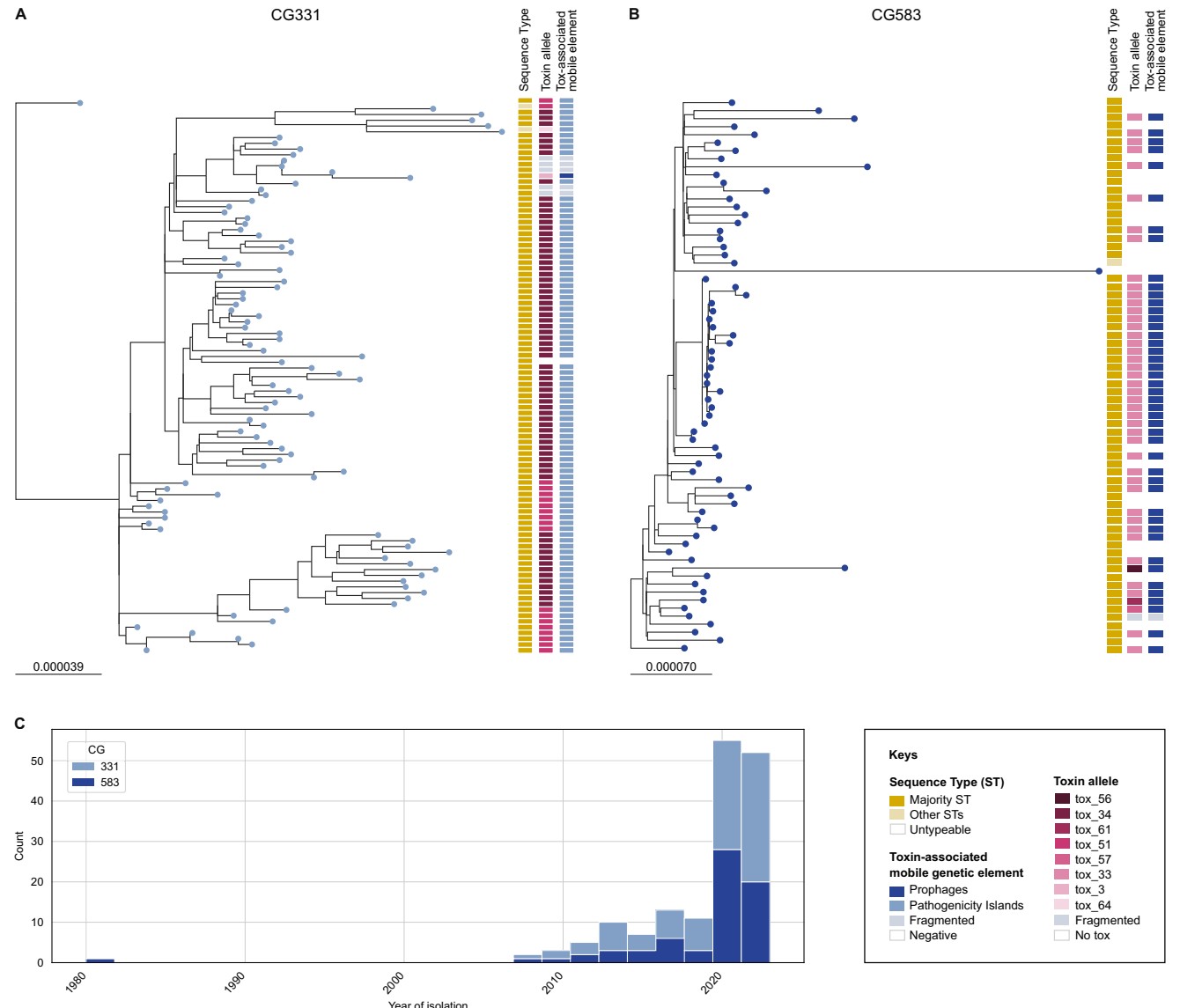

**Fig. 2 | Phylogenetic trees and isolation timeline of two major *Corynebacterium ulcerans* clonal groups (CG331, CG583).** The two maximum-likelihood phylogenetic trees were inferred from CG331 (**A**) and CG583 (**B**) isolates and rooted based on the complete *C. ulcerans* phylogenetic tree (Fig. 1). The first external strip shows the majority Sequence Type (ST) in dark yellow (ST331 for CG331, and ST325 for CG583) and other STs in lighter yellow (ST889 and ST933 for CG331, and ST583 and ST782 for CG583). The second and third external strips show the toxin alleles and the toxin-associated mobile genetic elements, respectively; **C** Timeline of isolation of CG331 and CG583 isolates.

differences, whereas case clusters had less than 20. However, the majority (*n* = 16) of the cryptic clusters show a genetic heterogeneity comparable to that of case clusters (e.g., <20 allele mismatches among all isolates). This strongly differs from what was observed previously in *C. diphtheriae*, in which most cryptic clusters show a higher heterogeneity compared to case clusters[21]. Cryptic clusters with low genetic heterogeneity in *C. ulcerans* could correspond to true case clusters for which there is simply a lack of evidence for epidemiological links, suggesting the utility of cgMLST for identifying possible undetected transmission.

Cryptic clusters are either limited to a single host species (*n* = 20; of which *n* = 12 human-only, *n* = 8 animal-only) or involve multiple hosts (*n* = 18). Among the latter, 10 include isolates from humans and pets (dogs, cats, and horses), whereas eight include only animal isolates from different species (both domestic and wildlife). Interestingly, no cluster includes both human and wild animal isolates, supporting the role of pets as the main source of infection for humans and as

transmission intermediates between humans and wildlife, which appears as the major reservoir of infection for domestic animals.

We also detected previously unreported links between two cryptic clusters and known case clusters. Both involve isolates that were collected in the same geographical areas as the epidemiologically related isolates, with a maximum distance of 60 km between isolation sites and a time range of six years for the first cluster, and of 50 km and nine years for the second cluster (Supplementary Fig. S3). One additional isolate from a homeless patient is linked to a known case cluster of two domestic rats from a nearby region that had been purchased from the same pet store[11] (Supplementary Fig. S3).

All cryptic clusters are limited to a single country, and most originate from France (*n* = 31), a direct consequence of the higher availability of genomes from the French NRC. Interestingly, three among the major cryptic clusters (FR_cryptic_02, *n* = 19 isolates; FR_cryptic_23, *n* = 7 isolates; and FR_cryptic_27, *n* = 8) show a strong association with specific geographical areas. The first was uniquely reported in the

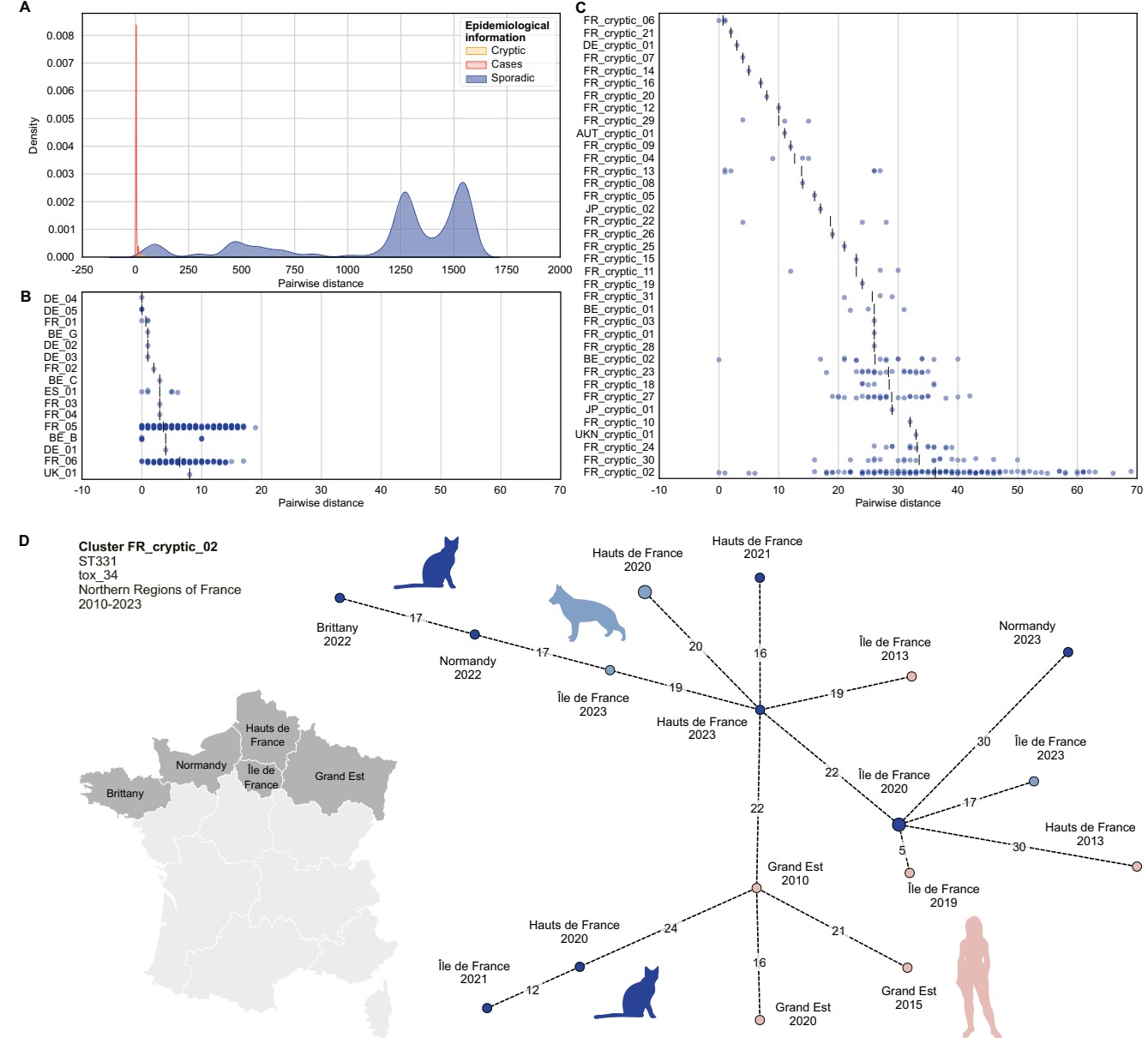

**Fig. 3 | Genetic heterogeneity of *Corynebacterium ulcerans* isolates from sporadic isolates, case clusters, and cryptic clusters. A** Density plot showing the pairwise distances (i.e., number of allelic differences) between core genome multilocus sequence typing (cgMLST) profiles among sporadic isolates, cryptic clusters and case clusters; **B** Pairwise distances within each case cluster (*n* = 16), excluding those with only one genome available (Supplementary Table S1); **C** Pairwise distances within each cryptic cluster (*n* = 37, excluding FR_cryptic_17, Fig. S3), defined as single-linkage groups with ≤25 allele mismatches and no associated epidemiological data. In (**B** and **C**), each dot represents the pairwise distance between two isolates, whereas each vertical black line indicates the mean for each cluster. **D** The major cryptic cluster detected in this work (FR_cryptic_02, sequence type ST331), shows a strong geographical association with the Northern regions of France and temporal spread with progressive dissemination to neighboring regions over time.

Northern regions of France (Fig. 3D), the second in the North-East, and the third was mostly found in the South (Supplementary Fig. S4). The spatial and temporal spread of these genetic clusters suggests progressive dissemination to neighboring regions over time, possibly through wildlife movement.

Taken together, these results suggest that carriage of *C. ulcerans* in wild animal populations plays an important role in its geographical spread (Fig. 3C, Supplementary Fig. S4), with pets likely being the major link between wildlife and humans. However, when looking at the allelic distances between available isolates within the cryptic clusters of transmission (as well as at their years of isolation and geographical origin), it is clear how a lot of the genetic diversity of the *C. ulcerans* population is currently not being captured. This reflects the severe

under-sampling, under-reporting, and under-sequencing of isolates from the wildlife population, which currently impairs our understanding of the transmission and phylodynamics of *C. ulcerans*.

## Diphtheria toxin gene variation in the *C. diphtheriae* Species Complex and within *C. ulcerans*

The diversity and evolutionary dynamics of the *tox* gene within the CdSC have been scarcely studied so far[23–25,31]. We therefore explored the diversity of the *tox* gene and the distribution of its variants across lineages and species. On a total of 582 *C. ulcerans* genome assemblies, 68.6% (*n* = 399) carry a complete *tox* gene, whereas for 13 additional *tox*+ genomes, only partial genes were detected (*tox* sequences straddling two contigs). Twenty new toxin alleles were detected from

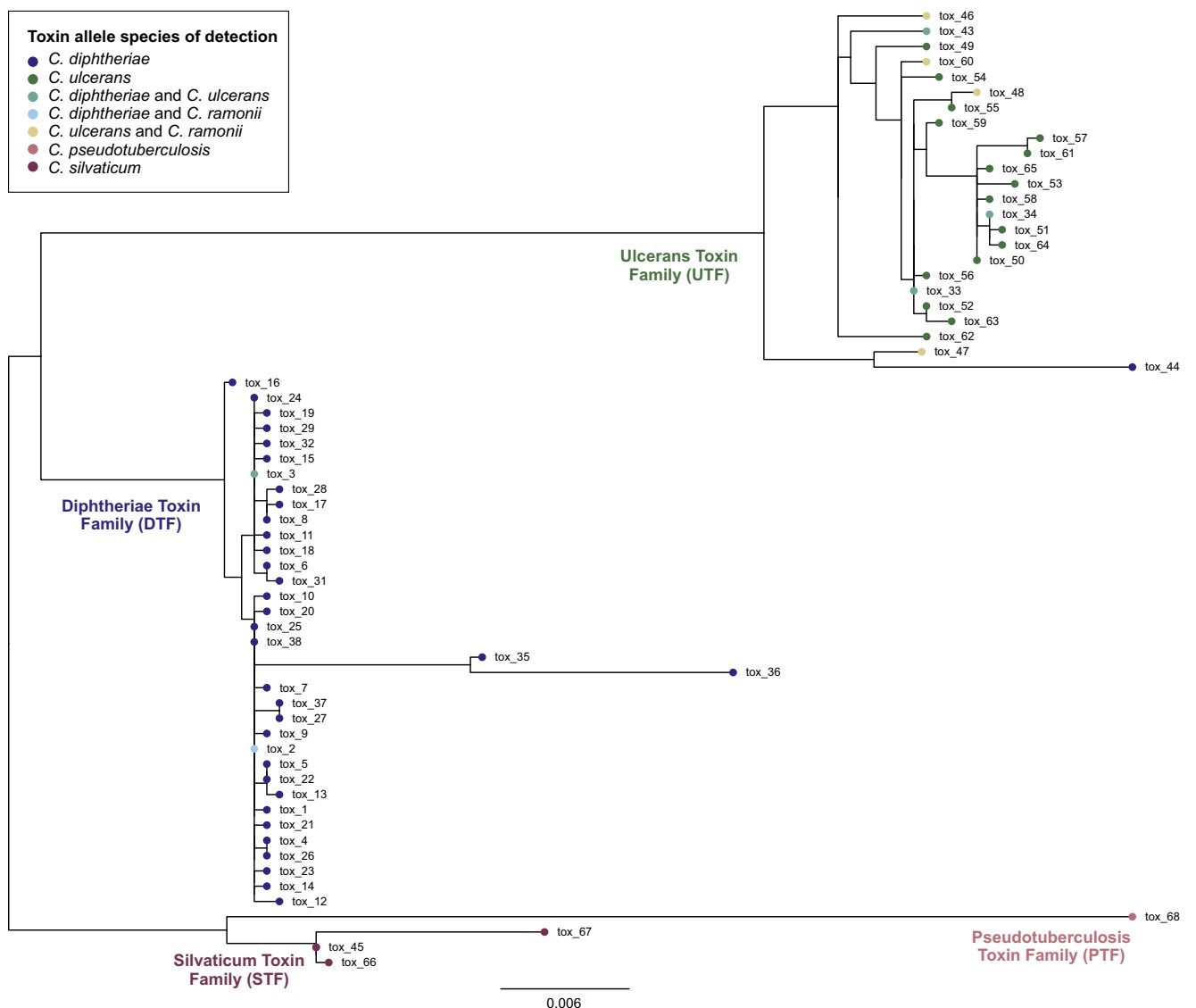

**Fig. 4 | Toxin families of the *Corynebacterium diphtheriae* Species Complex (CdSC).** This maximum-likelihood phylogenetic tree (midpoint rooting) was inferred from all toxin alleles gathered from every tox-carrying species of the CdSC: *C. diphtheriae*, *C. ulcerans*, *C. ramonii*, *C. silvaticum*, and *C. pseudotuberculosis*. Toxin alleles cluster into four main toxin families, named after their main bacterial host species: Diphtheriae Toxin Family (DTF; dark blue), Ulcerans Toxin Family (UTF; dark green), Silvaticum Toxin Family (STF; dark purple), and Pseudotuberculosis Toxin Family (PTF; light purple). Some alleles are unique to one bacterial host species (e.g., STF, PTF, and the majority of DTF alleles), whereas others are found in more than one species (e.g., DTF tox_2 and tox_3, and eight alleles of the UTF).

*C. ulcerans* genomes and added to the *tox* gene sequence allele definitions in the BIGSdb-Pasteur database (allele identifiers from tox_46 to tox_65).

The diversity of the diphtheria *tox* gene was compared across all members of the CdSC (*n* = 63 alleles in total; BIGSdb-Pasteur database, accessed September 1st, 2024), and four families (here named toxin families) were identified and labeled based on the main bacterial species in which they were observed (Fig. 4): Diphtheriae Toxin Family (DTF), Ulcerans Toxin Family (UTF), Silvaticum Toxin Family (STF), and Pseudotuberculosis Toxin Family (PTF). Previous studies reported a division in species-specific clades of their diphtheria toxin sequences[24,25], corresponding to these four families. However, in contrast to previous findings[24,25], here we show how some *tox* alleles are not unique to a single bacterial species (Fig. 4) or sublineage (Fig. 5A), likely as a result of horizontal gene transfer via phage excision, infection, and lysogeny (see below). Interestingly, a high proportion of UTF alleles (33.3%) were detected in bacterial species other

than *C. ulcerans* (Fig. 4). This differs from what was observed for the other toxin families (e.g., only 5.7% DTF alleles were detected in species other than *C. diphtheriae*), therefore suggesting a broader host range of *tox*-prophages from *C. ulcerans*, and a role for *C. ulcerans* as a reservoir of *tox* gene diversity that can be transmitted to other species, including *C. diphtheriae*.

## Mobile genetic elements associated with the diphtheria toxin gene

To better define the *tox*-carrying genetic elements in *C. ulcerans*, we first sought to clarify the location of orphan toxin genes (108/399, 27.1% of all *tox*+ isolates) in CG325 using supplementary long-read sequencing data (see "Methods" section). Strikingly, the resulting high-quality hybrid assemblies revealed the presence of two identical copies of the *tox* gene (allele tox_49), one carried by a 35,770 bp prophage, and the other by an integron-like element (ILE, of 3230 bp). The ILE is inserted a few genes (~8000 bp) downstream of the prophage

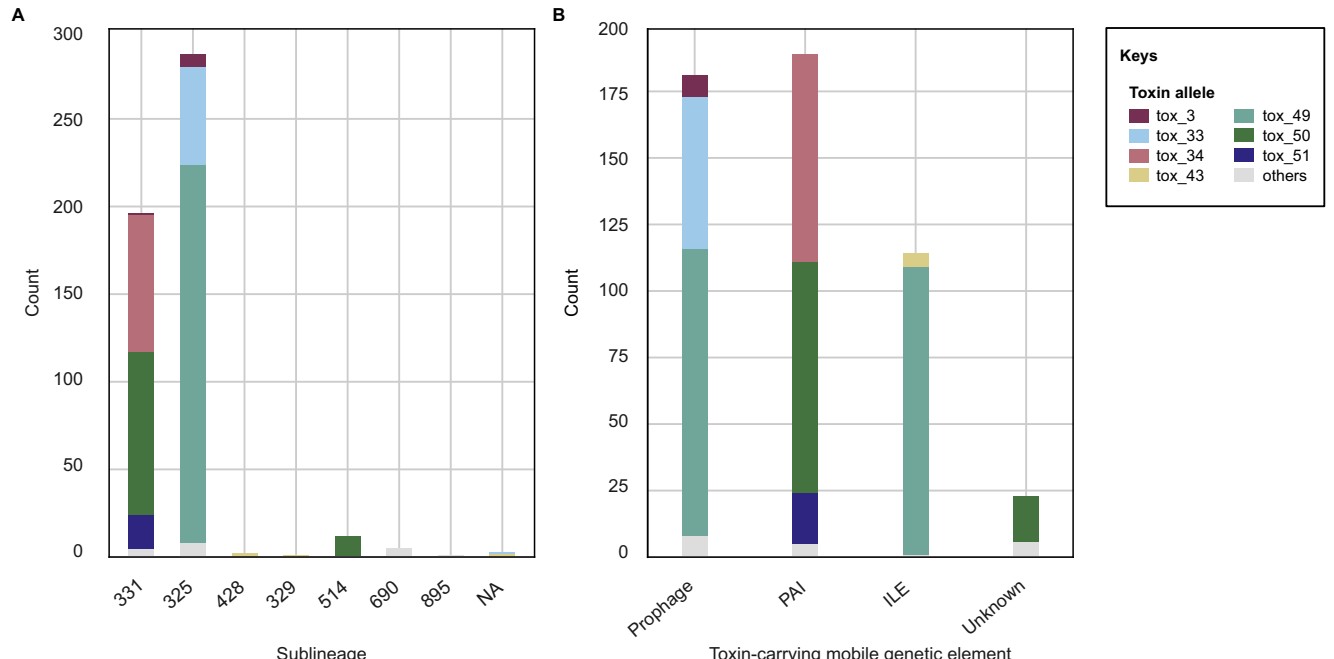

**Fig. 5 | Distribution of toxin alleles among sublineages and mobile genetic elements in *Corynebacterium ulcerans*. A** Bar plot showing the main tox alleles of *C. ulcerans* and their distribution among sublineages (SL); NA: not assigned. **B** Bar plot of the main tox alleles detected in *C. ulcerans* and the mobile genetic elements (MGE) carrying them; unknown MGE are due to high genome fragmentation; PAI pathogenicity island; ILE integron-like element.

(Supplementary Fig. S5), and its structure (constituted solely by a tyrosine recombinase/integrase, a hypothetical gene, and the toxin) is reminiscent of an integron, although IntegronFinder[32] did not identify it as such. The presence of two identical copies of the *tox* gene explains the failure of short-read-based genome assemblies, in which all *tox* gene reads from both copies were collapsed into a single contig. Carriage of two *tox* genes has so far only been observed in *C. diphtheriae*, in the toxin hyperproducer vaccine strain PW8. However, in PW8, these are both carried by two full prophages, a situation likely due to in vitro evolution upon strain selection for high-level toxoid vaccine production, as opposed to the natural occurrence observed here in *C. ulcerans*.

When also considering the other clonal groups (Fig. 1), the *tox* gene was approximately equally carried either by the PAI (189/399, 47.4% of the *tox*+ genomes, mostly in CG331) or by prophages (181/399, 45.4%, including the CG325 and CG583 prophages; Fig. 5B). In six genomes (1.5% of the *tox*+ isolates), the *tox* gene was carried uniquely on an ILE. Together with the CG325 genomes, the *tox*-ILE was present in 28.6% of *C. ulcerans* isolates.

Phylogenetic classification of the *tox*-carrying prophages reveals five main genetic groups, named here prophage families (PF1 to PF5; Fig. 6). Four prophage families are associated uniquely with one bacterial host species (PF2 with *C. diphtheriae*, and PF3-PF5 with *C. ulcerans*). In contrast, among PF1 prophages (primarily detected in *C. diphtheriae*), a subgroup can be observed in *C. ulcerans*. The inclusion of the *C. ulcerans* subgroup within the broader diversity of *C. diphtheriae* PF1 prophages suggests horizontal transmission of the PF1 bacteriophage from the latter to *C. ulcerans* (isolates belonging to both SL325 and SL331, suggesting two independent transfers). Prophage families unique to *C. ulcerans* (PF3- PF5) reflect a broad prophage diversity within this species but with primary association with SL325 (Supplementary Fig. S6). In contrast, PF1 and PF2 are found across several sublineages within *C. ulcerans* and/or *C. diphtheriae* (Supplementary Fig. S6), suggesting an ability to infect a broader range of strains.

Despite the evidence for horizontal transmission of bacteriophages, vertical transmission of *tox*-carrying elements seems to be

the predominant source of the *tox* gene in *C. ulcerans*. Ancestral acquisition of both *tox*-prophages and of the PAI is evident from their distribution in the *C. ulcerans* phylogenetic tree (Fig. 1): most *tox*+ strains share an ancestor that had probably already acquired a prophage (SL325) or a PAI (SL331). In addition, the PAI is well conserved across all clonal groups of SL331 (only $n = 740$ Single Nucleotide Polymorphisms across the 1,348,704 sites of the 189 aligned PAI sequences). The PAI does not seem to be excised or lost by other mechanisms, which differs from prophages that appear to have been excised multiple times from SL325 genomes. It is possible that the infection of multiple prophages in SL325 creates a competition leading to instability of the *tox* gene presence.

### The diversity of integrases in *tox*-prophages and other *tox*-carrying mobile elements

To provide insights into the mechanisms of integration of *tox*-carrying elements, we analyzed the diversity of their integrases, which determine the integration site[33,34]. The prophages and the ILE show diverse integrase amino-acid variants, whereas the PAI harbors a unique integrase sequence (Supplementary Fig. S7). Three integrase types were detected among *tox*-prophages, two of which were found in both *C. ulcerans* and *C. diphtheriae* (type 1 in PF2, PF3, and PF5; type 2 in PF1 and PF2; Fig. 6), and one uniquely in *C. ulcerans* (type 3 in PF4). Although phage integrase types 1 and 2 have different insertion sites (Supplementary Fig. S7), they show a high amino-acid sequence similarity (90%), whereas phage integrase 3 is quite distinct from the others (only ~33% sequence identity). The most common prophages in *C. ulcerans* are those with integrase 1 (Fig. 6), and they share the same insertion site with the PAI (Supplementary Fig. S7). This likely explains why these elements appear as mutually exclusive in their respective phylogenetic branches.

These findings open interesting questions about the co-evolution of the diphtheria *tox* gene, its associated mobile genetic elements, and their carrying strains. Considering the introduction of the toxoid vaccine against diphtheria in the 1920s (*tox* allele 2, DTF family), it could be expected that vaccine-driven immunity might have

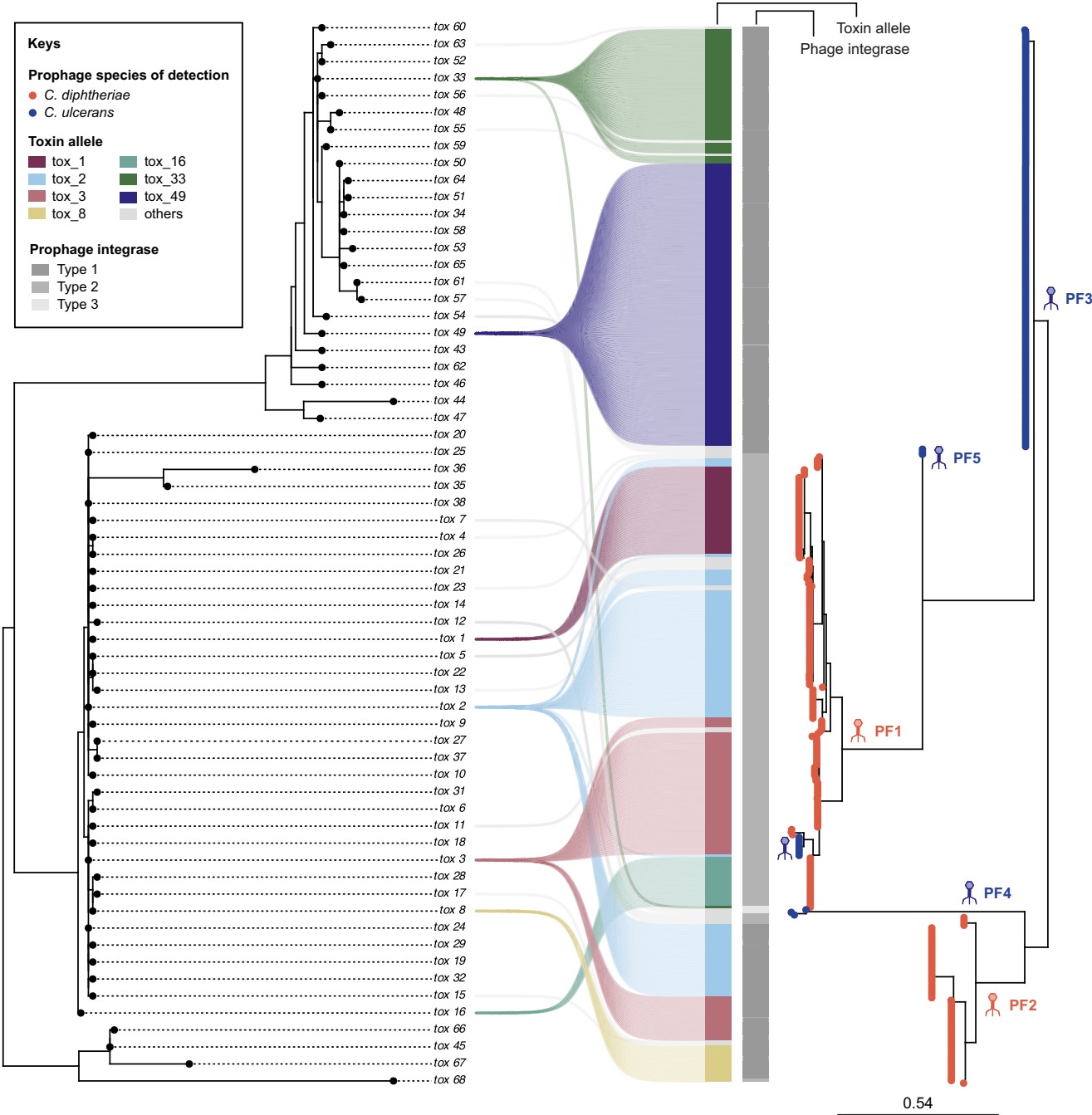

**Fig. 6 | Toxin alleles of the *Corynebacterium diphtheriae* Species Complex (CdSC) and their association with *tox*-prophage families.** Phylogenetic classifications of *C. ulcerans* (*n* = 176) and *C. diphtheriae* (*n* = 232) *tox*-prophages (right) and their *tox* alleles (left), with main alleles shown in color. Prophage families (PF) are indicated on the internal branches of the corresponding subtree in the prophage phylogenetic tree (PF1–PF5), in which leaves are colored according to the bacterial host species of detection. The phage integrase type (1–3) is depicted on the right-hand side of the *tox* allele as shades of gray.

disadvantaged primarily the DTF family and prophages from PF1 and PF2. However, despite several differences at the amino-acid level of the diphtheria toxin from *C. diphtheriae* and *C. ulcerans* in the receptor-binding domain and in the translocation region[23], the diphtheria toxoid vaccine is also active against the diphtheria toxin produced by *C. ulcerans*[35]. Considering the current epidemiological emergence of *C. ulcerans* cases worldwide, the possible impact of the diphtheria vaccine on *tox*-carrying *C. ulcerans* strains and mobile genetic elements is an important question. The data presented here could help frame future priority research questions, such as what is the effect of vaccine-driven selective pressure on UTF variants and on prophage families from *C. ulcerans* (PF3-PF5), or whether acquisition of a *tox* allele from

the UTF via phage infection (e.g., PF1 prophages carrying tox_3; Fig. 6) affects the pathogenicity of *C. ulcerans*, and vice versa (e.g., tox_33; Fig. 4).

### Limitations
Availability of *C. ulcerans* genomic data from outside Europe is scarce (see Supplementary Data 1), and the picture provided by this work is influenced by a bias towards isolates from France, a direct consequence of the large collection of isolates sequenced in the frame of the French surveillance of diphtheria. In addition, current under-sampling of isolates from wild animal host species does not allow to fully capture the broader diversity of the *C. ulcerans* population.

This work provides a comprehensive analysis of the population structure, genomic epidemiology, and toxin gene diversity of the zoonotic pathogen *C. ulcerans*. Using a dedicated cgMLST scheme made openly accessible, prominent sublineages and clonal groups were identified, revealing geographic spread and host distribution, and uncovering potential transmission chains. Our study highlights the role of pets in zoonotic transmission and calls for enhanced surveillance, especially of domestic and wild animal isolates. It also provides an updated overview of the diphtheria toxin gene diversity, confirming the demarcation of four toxin families[25]. Additionally, it reports the first *tox*-carrying integron-like element, marking the discovery of a naturally occurring double-*tox*+ *C. ulcerans* clone, and reveals how *tox*-carrying mobile genetic elements are involved in both vertical inheritance and horizontal gene transfer. This improved understanding of *C. ulcerans* diversity and evolution provides a population genomic framework that will enhance future epidemiological tracking and infection control.

## Methods

### Bacterial isolates and case clusters included in the study

Three datasets were used for this study: (i) an original dataset comprising 347 *C. ulcerans* genome assemblies available in June 2022 ($n = 317$ from the French NRC and the Collection of Institut Pasteur and $n = 30$ from GenBank); (ii) an expanded dataset of 434 genome assemblies, comprising additional *C. ulcerans* ($n = 51$) and *C. ramonii*[36] ($n = 36$) isolates from private and public repositories; (iii) a validation dataset of 582 *C. ulcerans* genome assemblies, comprising additional ones ($n = 184$) from the French NRC and from the diphtheria BIGSdb database[37]. A complete list with relative metadata can be found in Supplementary Data 1.

The first two datasets (original and expanded) were used to build a cgMLST scheme for *C. ulcerans* that would also be applicable to *C. ramonii*, as these species are phylogenetically closely related[36]. The third dataset was used for: (i) validating the cgMLST scheme; (ii) defining sublineages and clonal groups; (iii) exploring case clusters ($n = 20$, Supplementary Table S1) and clusters with cryptic transmission.

### Isolate growth and DNA extraction

Isolates were grown in Tryptic Soy Agar (TSA) for 24–48 h at 35–37 °C and then resuspended in sterile saline (0.9% NaCl) for DNA extraction (optical density, OD = 1), which was performed using the DNeasy Blood & Tissue Kit (QIAGEN, Hilden, Germany). A lysis step was added to the extraction protocol described by the manufacturer[38].

DNA extraction for Oxford Nanopore sequencing was performed on a Maxwell RSC Instrument (Promega, Madison, USA) with the Maxwell RSC Blood DNA Kit (Promega, Madison, Wisconsin, USA) following the manufacturer's instructions.

### Genomic sequencing and assembly

Whole genome short-read sequencing was performed from Nextera XT libraries on Illumina NextSeq 500 and NextSeq 2000 apparatuses (Illumina, San Diego, USA) with a 2 × 150 nt paired-end protocol. Paired-read sequence data were assembled using fq2dna v21.06 (https://gitlab.pasteur.fr/GIPhy/fq2dna). ST determination and occurrence of the *tox* gene were performed using diphtOscan[39].

Long-read sequencing was carried out using Oxford Nanopore MinION R9.4.1 flow cells with the Rapid Barcoding kit (SQK-RBK004). Basecalling was performed using Guppy v6.4.2 (SUP algorithm), and high-quality hybrid Illumina/Nanopore assemblies were obtained using Unicycler v0.4.8[40].

### Definition of the cgMLST scheme

The cgMLST scheme was built on the first two genomic datasets (original and expanded) using chewBBACA v2.8.5[41], and refined by applying specific criteria[21] (for a detailed description, please refer to the Supplementary Information file). The final scheme comprises 1628 loci and is available in the BIGSdb-Pasteur diphtheria database (https://bigsdb.pasteur.fr/diphtheria).

### Profile-based population structure analysis

To identify optimal thresholds of allelic mismatch thresholds among cgMLST profiles that would define genetic clusters most-fitting to the *C. ulcerans* population structure, we used MSTclust (https://gitlab.pasteur.fr/GIPhy/MSTclust) to compute the consistency (silhouette) coefficient of the single-linkage clustering groups derived from every allele mismatch threshold value (from 1 to 1628) in order to determine those corresponding to local and global optima[29] (Supplementary Fig. S1).

### Diphtheria toxin gene and *tox*-associated mobile elements

The diphtheria *tox* gene was detected on available isolates by a qPCR assay[38], and the production of the toxin was evaluated with the modified Elek immunoprecipitation test[42].

The presence of the *tox* gene and of other *C. ulcerans*-specific virulence genes (e.g., *pld, rbp*) was assessed in the validation dataset with diphtOscan[39]. When a new *tox* allele was found, the nucleotide sequence was added to the toxin allele database in BIGSdb-Pasteur.

All *tox*+ genomes were scanned using blastn[43] for the presence of the PAI (accession KP019622.1), and PHASTER[44] for the presence of prophages. Orphan toxins (i.e., present as a single gene contig) were detected consistently among CG325 genomes; therefore, three isolates from this clonal group (FRC1215, FRC1277, FRC1308) were selected for long-read sequencing in order to clarify the location of the toxin. To reconstruct the *tox*-prophage from the remaining genomes of the CG325, the short raw reads were aligned to the reference prophage of FRC1215 using bwa-mem2 v2.2.1[45], to next derive consensus sequences using samtools v1.18[46].

### Phylogenetic analyses

A core gene alignment (2101 loci, computed using Panaroo v1.5.0[47], default settings) was used to infer a phylogenetic tree using IQ-TREE v2.0.6 (evolutionary model GTR + F + R8)[48,49]. Four reference *Corynebacterium pseudotuberculosis* genomes from reference strains (ATCC19410, CCUG27541, NCTC4656, NCTC4681) were used as an outgroup. Bootstrap values can be visualized within Microreact[50] (see project number in the "Data availability" section). Tree files in svg format were exported from Microreact, and figures were edited using Inkscape (available from: https://inkscape.org).

Multiple nucleotide sequence alignments were computed for (i) all *tox* alleles in the BIGSdb database ($n = 63$), (ii) full prophage sequences ($n = 176$ from *C. ulcerans* and $n = 232$ from *C. diphtheriae*), and (iii) the PAI ($n = 189$) using MAFFT v7.526[51] for the former (i), and Clustal-Omega v1.2.4[52] for the latter two (ii–iii; default settings). Integrase genes were extracted from prophage and PAI sequences using tblastn. Phylogenetic trees for both the *tox* alleles and the corresponding *tox*-prophages were reconstructed from the multiple sequence alignments using FastTree v2.1.11[53] using the Jukes–Cantor model (default parameters). Phytools v2.1-1[54] was used to obtain a co-phylogeny of the toxin alleles and the prophage sequences.

### Reporting summary

Further information on research design is available in the Nature Portfolio Reporting Summary linked to this article.

## Data availability

Supplementary material for this work includes a Supplementary Information file and the Supplementary Data 1 file (which contains all

isolate metadata for strains used in this study, and results of genomic and laboratory analyses). The genome sequence data generated in this study have been deposited and are publicly available in the European Nucleotide Archive (ENA); all accession numbers can be found in Supplementary Data 1. Additionally, genome sequences used in this study were made available at BIGSdb-Pasteur (project id 45 at https://bigsdb.pasteur.fr/cgi-bin/bigsdb/bigsdb.pl?db=pubmlst_diphtheria_isolates). A project with the phylogenetic tree, metadata, and results file is available within Microreact at: https://microreact.org/project/pJf35xQkcs2HUznD67W7Be-corynebacterium-ulcerans-genomic-epidemiology-2025. Source data are provided with this paper.

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

## Acknowledgements
The NRC for Corynebacteria of the *diphtheriae* complex is supported financially by Institut Pasteur and Santé Publique France (Public Health France). This work was supported financially by the French Govern-ment's Investissement d'Avenir grant Laboratoire d'Excellence Inte-grative Biology of Emerging Infectious Diseases (ANR-10-LABX-62-IBEID). We thank Dr Julie Toubiana for her clinical expertise and support to the French NRC. We thank the Mutualized Platform for Microbiology (P2M, Institut Pasteur) for sequencing isolates using Illumina technology. This work used the computational and storage services provided by the IT Department at Institut Pasteur.

## Author contributions
Conceptualization: Sylvain Brisse, Chiara Crestani. Methodology, Data Curation, Validation, and Visualization: Chiara Crestani. Investigation: Nora Zidane, Virginie Passet, Sylvie Brémont, Edgar Badell. Software: Martin Rethoret-Pasty, Alexis Criscuolo. Formal Analysis: Chiara Cres-tani, Alexis Criscuolo. Writing original draft: Chiara Crestani. Writing Review and Editing: Sylvain Brisse, Chiara Crestani, Alexis Criscuolo. Supervision and Funding Acquisition: Sylvain Brisse.

## Competing interests
The authors declare no competing interests.
