## [Peer Review file · Nature Communications]

Microevolution and genomic epidemiology of the diphtheria-causing zoonotic pathogen *Corynebacterium ulcerans*

Corresponding Author: Professor Sylvain Brisse

Version 0:

Reviewer comments:

Reviewer #1

(Remarks to the Author)

The manuscript submitted by Crestani et al. provides a systematic analysis of *Corynebacterium ulcerans* microevolution and tox gene carrying genetic elements based on almost 6 hundred genome sequences.

The manuscript is well-written. A few points may be addressed to improve the impact of the paper:

throughout the text: *tox+* and *tox-* are phenotypic descriptions and should not be in italics

Figure 1: text within the figure is difficult to read, either use black color instead of white or redesign the figure.

Figure 3: grey letters and numbers difficult to read

Figure S3 may be included in the manuscript

Reviewer #2

(Remarks to the Author)

The study explores the species *Corynebacterium ulcerans*, part of the diphtheriae complex, through population genomics. The contextualization presented in the introduction and discussion is well-structured, supported by up-to-date references. The methodology is modern and comprehensively examines the structure and phylogenomic relationships of *C. ulcerans*, correlating these findings with data on the isolation site, cgMLST allelic profiles, the *tox* gene associated with transposable elements such as pathogenicity islands and prophages, as well as the host.

There are a few considerations that should be taken into account. For instance, regarding the choice of software for identifying genomic islands, there are more recent programs that could be explored in addition to BLAST for this purpose. In the section on the "Population Structure of *C. ulcerans*," it appears that only *C. ulcerans* strains were analyzed. To draw more robust conclusions, it would be necessary to include comparisons with *C. diphtheriae*. Furthermore, although the construction of phylogenomic trees was mentioned, bootstrap values were not provided, which are essential for statistically demonstrating the reliability of these trees.

The images and their correlations are very well presented. However, I believe the methodology section could be improved by providing a more detailed description of how these images were generated, including whether any specific software or packages were used.

Reviewer #3

(Remarks to the Author)

Corynebacterium ulcerans, a member of the *Corynebacterium diphtheria* species complex (CdSC), is a bacterium of concern to veterinary public health (VPH). It is known to be transmitted from companion animals (in particular) causing infections in susceptible humans. Little to no information describing the genomic structure of clinically relevant clones expressing a diphtheria-positive toxin phenotype, in respect of this zoonotic pathogen has appeared in the literature. Consequently

transmission pathways remains to be elucidated. This paper set out to describe the genomic population structure of *Corynebacterium ulcerans* commenting on its evolution from a study collection of 582 isolates, taken from a range of different geographical locations and hosts. A core genome-based typing scheme was developed and evaluated and facilitating the early deciphering of transmission routes between companion animals and humans. Furthermore a detailed characterisation of the genomic nature of the associated diphtheria toxin-encoding gene (*tox*) was also presented and this work established the occurrence of three broad representations of this marker, including within four diphtheria toxin families [named according to their species of origin]; five *tox*-prophage families and a novel integron-like arrangement. This work also demonstrated evidence of cross-species transduction, highlighting the fact that some of these broad groupings are shared across the *Corynebacteria* species..

What are the noteworthy results?

As mentioned above, this study is of importance to VPH as it provides some useful new tools that when applied, and which extend our understanding of this zoonotic pathogens molecular epidemiology. Specifically a standardised genotyping protocol, based on the classical cgMLST strategy (in this case comprising some 1,628 highly conserved loci), was developed, capable of providing useful clinical information by differentiating between *C. ulcerans*. In addition detailed characterisation of the diphtheria toxin-encoding gene provides useful information describing the diversity of *tox*-carrying mobile genetic elements (MGE) and the occurrence of variants.

Will the work be of significance to the field and related fields? How does it compare to the established literature? If the work is not original, please provide relevant references.

Undoubtedly this work is important delivering a well-designed approach to the description of a hitherto underreported zoonotic bacterial pathogen of importance to VPH. Data described lends itself to the creation of a framework by which surveillance of members of the CdSC can be considered in a One Health context can be considered.

Does the work support the conclusions and claims, or is additional evidence needed?

Based on this reviewers thorough assessment of the work presented, the conclusions drawn are scientifically sound. As stated above the technical approach used to describe the molecular epidemiology of CdSC is valid.

Are there any flaws in the data analysis, interpretation and conclusions? Do these prohibit publication or require revision?

None that can be identified. The consideration of the data presented appears to be thorough and described in a measured manner, with due care and attention.

Is the methodology sound? Does the work meet the expected standards in your field?

As stated earlier and avoiding the risk of repetition, the technical methodology is sound. It certainly is a study of the highest quality that will be of interest to a very broad readership, focused on VPH and beyond.

Is there enough detail provided in the methods for the work to be reproduced?

As above.

This is a most interesting and relevant body of work aimed at describing the molecular epidemiology of *C. ulcerans*, a hitherto underrecognized zoonotic bacterial pathogen. It established the role of companion animals and reservoirs of the infectious agents and their role in transmission. Further the data presented provides novel insights into the emergence and dissemination of the *tox* gene and considers the appropriateness or otherwise of the current vaccination measures. ,

Minor items that should be addressed -

L288- delete the word by;

L423- used to...;

ANSWERS TO REVIEWER COMMENTS

Reviewer #1 (Remarks to the Author):

The manuscript submitted by Crestani et al. provides a systematic analysis of *Corynebacterium ulcerans* microevolution and tox gene carrying genetic elements based on almost 6 hundred genome sequences.

The manuscript is well-written. A few points may be addressed to improve the impact of the paper:

throughout the text: tox+ and tox- are phenotypic descriptions and should not be in italics

Figure 1: text within the figure is difficult to read, either use black color instead of white or redesign the figure.

Figure 3: grey letters and numbers difficult to read

Figure S3 may be included in the manuscript

We thank the reviewer for appreciating our work, and for the helpful comments. Below, we address each point raised:

- Toxin gene nomenclature – In our manuscript, we chose to follow the convention that utilizes italics to denote the toxin gene's nomenclature (*tox*), distinguishing it from phenotypic descriptions (toxigenic strains). The nomenclatures “tox+” and “tox-” only refer to gene presence/absence, as detailed in lines 109-113.
- Figure 1 – we have now substituted the white text with dark grey text to improve readability
- Figure 3 – we have now substituted the white and grey text with black text to improve readability

Reviewer #2 (Remarks to the Author):

The study explores the species *Corynebacterium ulcerans*, part of the diphtheriae complex, through population genomics. The contextualization presented in the introduction and discussion is well-structured, supported by up-to-date references. The methodology is modern and comprehensively examines the structure and phylogenomic relationships of *C. ulcerans*, correlating these findings with data on the isolation site, cgMLST allelic profiles, the tox gene associated with transposable elements such as pathogenicity islands and prophages, as well as the host.

There are a few considerations that should be taken into account. For instance, regarding the choice of software for identifying genomic islands, there are more recent programs that could be explored in addition to BLAST for this purpose. In the section on the "Population Structure of **C. ulcerans**, " it appears that only **C. ulcerans** strains were analyzed. To draw more robust conclusions, it would be necessary to include comparisons with **C. diphtheriae**. Furthermore, although the construction of phylogenomic trees was mentioned, bootstrap values were not provided, which are essential for statistically demonstrating the reliability of these trees.

The images and their correlations are very well presented. However, I believe the methodology section could be improved by providing a more detailed description of how these images were generated, including whether any specific software or packages were used.

We thank the reviewer for taking the time to thoroughly read our work and to provide constructive feedback. While we acknowledge the existence of more recent software for detecting genomic islands, we specifically aimed to focus on a known pathogenicity island with an established reference sequence (Dangel et al., 2019; lines 60-61), which rendered BLAST the most suitable choice for our analysis. With regards to the section “Population structure of *C. ulcerans*”, indeed we uniquely analyzed genomes of this species (and of its sister species *C. ramonii*) in this study. However, in this section, we do include comparisons

with the population structure of *C. diphtheriae*, based on previous studies (lines 89-93; Guglielmini et al., 2021; Hennart et al., 2020).

Bootstrap values for phylogenetic analyses can be visualized directly in the Microreact project (<https://microreact.org/project/pJf35xQkcs2HUznD67W7Be-corynebacterium-ulcerans-genomic-epidemiology-2024>), by selecting the "internal labels" option in the menu "Nodes & labels". We have added this information in the methods section (lines 426-427). Finally, with regards to the methodology for figure generation and processing, details can be found at lines 427-429.

Reviewer #3 (Remarks to the Author):

Corynebacterium ulcerans, a member of the *Corynebacterium diphtheria* species complex (CdSC), is a bacterium of concern to veterinary public health (VPH). It is known to be transmitted from companion animals (in particular) causing infections in susceptible humans. Little to no information describing the genomic structure of clinically relevant clones expressing a diphtheria-positive toxin phenotype, in respect of this zoonotic pathogen has appeared in the literature. Consequently transmission pathways remains to be elucidated. This paper set out to describe the genomic population structure of *Corynebacterium ulcerans* commenting on its evolution from a study collection of 582 isolates, taken from a range of different geographical locations and hosts. A core genome-based typing scheme was developed and evaluated and facilitating the early deciphering of transmission routes between companion animals and humans. Furthermore a detailed characterisation of the genomic nature of the associated diphtheria toxin-encoding gene (*tox*) was also presented and this work established the occurrence of three broad representations of this marker, including within four diphtheria toxin families [named according to their species of origin]; five *tox*-prophage families and a novel integron-like arrangement. This work also demonstrated evidence of cross-species transduction, highlighting the fact that some of these broad groupings are shared across the *Corynebacteria* species..

What are the noteworthy results?

As mentioned above, this study is of importance to VPH as it provides some useful new tools that when applied, and which extend our understanding of this zoonotic pathogens molecular epidemiology. Specifically a standardised genotyping protocol, based on the classical cgMLST strategy (in this case comprising some 1,628 highly conserved loci), was developed, capable of providing useful clinical information by differentiating between *C. ulcerans*. In addition detailed characterisation of the diphtheria toxin-encoding gene provides useful information describing the diversity of *tox*-carrying mobile genetic elements (MGE) and the occurrence of variants.

Will the work be of significance to the field and related fields? How does it compare to the established literature? If the work is not original, please provide relevant references.

Undoubtedly this work is important delivering a well-designed approach to the description of a hitherto underreported zoonotic bacterial pathogen of importance to VPH. Data described lends itself to the creation of a framework by which surveillance of members of the CdSC can be considered in a One Health context can be considered.

Does the work support the conclusions and claims, or is additional evidence needed?

Based on this reviewers thorough assessment of the work presented, the conclusions drawn are scientifically sound. As stated above the technical approach used to describe the molecular epidemiology of CdSC is valid.

Are there any flaws in the data analysis, interpretation and conclusions? Do these prohibit publication or require revision?

None that can be identified. The consideration of the data presented appears to be thorough and described in a measured manner, with due care and attention.

Is the methodology sound? Does the work meet the expected standards in your field?

As stated earlier and avoiding the risk of repetition, the technical methodology is sound. It certainly is a study of the highest quality that will be of interest to a very broad readership, focused on VPH and beyond.

Is there enough detail provided in the methods for the work to be reproduced?

As above.

This is a most interesting and relevant body of work aimed at describing the molecular epidemiology of *C. ulcerans*, a hitherto underrecognized zoonotic bacterial pathogen. It established the role of companion animals and reservoirs of the infectious agents and their role in transmission. Further the data presented provides novel insights into the emergence and dissemination of the *tox* gene and considers the appropriateness or otherwise of the current vaccination measures. ,

Minor items that should be addressed -
L288- delete the word by;
L423- used to...;

We sincerely thank the reviewer for the positive feedback on our manuscript and for the suggestions. We have made the appropriate changes at the lines indicated.